# Modular Impulsive Green Monopropellant Propulsion System (MIMPS-G): For CubeSats in LEO and to the Moon

Ahmed E. S. Nosseir [1,2,*] , Angelo Cervone [1,*] and Angelo Pasini [2,*]

1 Department of Space Engineering, Faculty of Aerospace Engineering,
Delft University of Technology (TU Delft), 2629 Delft, The Netherlands
2 Sede di Ingegneria Aerospaziale, Dipt. di Ingegneria Civile e Industriale, Università di Pisa (UniPi),
56122 Pisa, Italy
* Correspondence: a.e.s.nosseir@student.tudelft.nl or a.nosseir@studenti.unipi.it (A.E.S.N.);
a.cervone@tudelft.nl (A.C.); angelo.pasini@unipi.it (A.P.)

**Abstract:** Green propellants are currently considered as enabling technology that is revolutionizing the development of high-performance space propulsion, especially for small-sized spacecraft. Modern space missions, either in LEO or interplanetary, require relatively high-thrust and impulsive capabilities to provide better control on the spacecraft, and to overcome the growing challenges, particularly related to overcrowded LEOs, and to modern space application orbital maneuver requirements. Green monopropellants are gaining momentum in the design and development of small and modular liquid propulsion systems, especially for CubeSats, due to their favorable thermophysical properties and relatively high performance when compared to gaseous propellants, and perhaps simpler management when compared to bipropellants. Accordingly, a novel high-thrust modular impulsive green monopropellant propulsion system with a micro electric pump feed cycle is proposed. MIMPS-G500mN is designed to be capable of delivering 0.5 N thrust and offers theoretical total impulse $I_{tot}$ from 850 to 1350 N s per 1U and >3000 N s per 2U depending on the burnt monopropellant, which makes it a candidate for various LEO satellites as well as future Moon missions. Green monopropellant ASCENT (formerly AF-M315E), as well as HAN and ADN-based alternatives (i.e., HNP225 and LMP-103S) were proposed in the preliminary design and system analysis. The article will present state-of-the-art green monopropellants in the (EIL) Energetic Ionic Liquid class and a trade-off study for proposed propellants. System analysis and design of MIMPS-G500mN will be discussed in detail, and the article will conclude with a market survey on small satellites green monopropellant propulsion systems and commercial off-the-shelf thrusters.

**Keywords:** green monopropellant; chemical rocket propulsion; CubeSats; small satellites; micro electric pump feed cycle



## 1. Introduction

CubeSat propulsion is evolving to fulfill the requirements of modern space missions and applications that demand propulsion capabilities to enable active orbital operations, such as orbital altitude and inclination changes, orbital transfers, formation flying, rendezvous operations–generally, operations requiring high-thrust impulsive maneuvers. An example for commercial CubeSats utilizing a green propulsion system, namely HPGP by ECAPS, is the SkySat LEO imaging constellation by Planet Lab from 2016 to 2020 [1]. Other science missions for CubeSats utilizing a propulsion system are MarCO Mars deep-space CubeSat utilizing a cold-gas propulsion system launched in May 2018 [2], and Pathfinder Technology Demonstrator (PTD) by NASA, launched in January 2021 which utilizes the Hydros-C water-based propulsion system [3]. Challenges facing this evolution include, as an example, the need for design-modularity and components miniaturization. Design modularity may be considered as a cornerstone for rapid fabrication and assembly of

subsystems and components, which usually reduce development costs and time. Miniaturization of components is crucial to the space industry in general, since nowadays every gram of payload mass to orbit may have a significant monetary value, adding to that the presence of onboard size restrictions. Design modularity and miniaturization is a major point of focus for various research work in the space propulsion field in general, either for electric propulsion [4,5] or for chemical propulsion in standalone systems or in multimode systems, as extensively studied by Rovey, J. L. et al. [6]. On another note, green monopropellants are the current trend in liquid propellant propulsion for small satellites, either in scientific or industrial research and development as well as commercial activities, due to their safety, stability, storability, relative design simplicity, as well as high performance, and may soon face global legal regulations for a greener environment–as expected by the authors. These facts were the motive behind the design of (MIMPS-G) the Modular Impulsive Propulsion System to utilize Green monopropellants and is a prospective system for micro and nano spacecraft, particularly CubeSats, requiring a modular propulsion system for high-thrust impulsive orbital maneuvers. From the study of the market and the current state-of-the-art products in the green propulsion industry, it was deemed necessary to design a green monopropellant propulsion system that would help in solving several challenges related to acquiring higher performances and lower costs as well as demonstrating competitive advantages to currently proposed systems, as will be discussed in Sections 4 and 5 of this manuscript. The design and development plans have taken place within a research work carried out at the beginning of the year 2020 between the Department of Aerospace Engineering in the University of Pisa and the Department of Space Engineering of the Aerospace Engineering Faculty in TU Delft.

The baseline design of MIMPS-G500mN is a standard 1U CubeSat size that can be expanded or clustered depending on the spacecraft size, required thrust level, and mission's $\Delta V$ budget. One of the critical components in this propulsion system that required special attention and deep analysis was the feed and pressurization system. It was found that conventional systems such as stored gas or blow-down pressure-fed systems were introducing more limitations over time, especially due to the increasing performance requirements and the size and mass restrictions on the inert parts of CubeSats. Foreseeing that eventually, designers would face design-simplicity trade-offs in favor of performance, it was time to investigate, study, and analyze unconventional and more complex feed and pressurization systems for small-sized spacecraft. Thus, in the preliminary design of MIMPS-G500mN, Figure 1 and Video S1, autogenous pressurization and micro electric pump feed (micro e-Pump feed cycle) were proposed using commercial off-the-shelf (COTS) components.

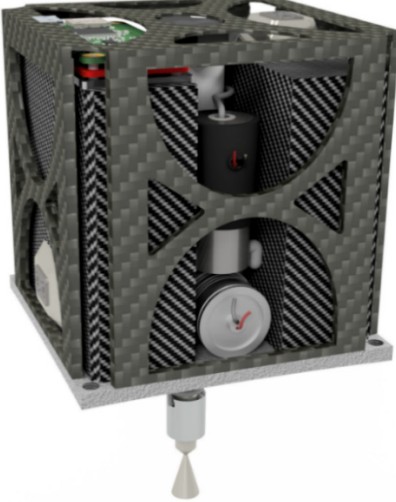

**Figure 1.** MIMPS-G500mN realistic render.

In the following sections of this article, the three proposed green monopropellants, belonging to the Energetic Ionic Liquids (EILs) class, will be reviewed emphasizing their physical properties, performance and their development status. These three selected propellants were a result of a trade-off study that will be discussed in detail in Section 1.2. Furthermore, the feed and pressurization systems of the designed propulsion system will be discussed, and the basic concepts will be elaborated on. The rest of the article will discuss the system analysis, requirements identification, design methodology, and preliminary design process and the results will be numerically tabulated. Finally, a market survey on the state-of-the-art monopropellant propulsion systems for small-sized spacecraft, as well as commercial off-the-shelf green monopropellant thrusters will be presented, highlighting the main performance parameters and technical specifications of such systems and thrusters to serve as a reference for our proposed propulsion system MIMPS-G500mN, as well as a reference for the readers of this manuscript. This article presents a more detailed analysis and results and extends the research work presented by the authors in the conference papers [7–9].

### 1.1. Space Mission Requirments

Spacecraft propulsion systems are typically designed and developed according to a predefined set of requirements dictated by the space mission analysis and design phase. Usually, any modification or compromise during the project development affects the design process and outcomes of the spacecraft's different systems and subsystems in order to maintain the strict requirements of the mission orbital operations. In addition to that, the size restrictions in micro- and nanosatellites inherit more challenges and limitations on the spacecraft systems' development, especially the propulsion system and its subsystems, which in turn leads to the development of a "single-purpose" or "one-time-use" micro propulsion systems that are solely developed for a particular mission.

To overcome such challenges, scientists and engineers are focused on optimizing various spacecraft component designs such as the power generation and storage systems, electronics, communication and control systems, and structural interfaces onboard the spacecraft to provide more integration flexibility and adaptability. The propulsion system remains one of the most challenging parts to optimize in terms of maintaining high performance and suitable costs. In the last two years, a lot of scientific efforts were put together in order to reach a new level of optimization through pushing the boundaries of systems engineering and bending the norms of conventional design and manufacturing as well as investigating new propulsion subsystems operation concepts. It was found that it is time to accept drastic changes and to consider trading off design simplicity for high-performance by manipulating current technologies to adapt more complex propulsion feed and pressurization systems as well as new propellant storage tank designs.

In the design of MIMPS-G, the greatest focus was put on modularity and expandability as key design elements to enable flexibility and adaptation of the propulsion system to various space mission requirements, especially the ones defined by modern orbital operations from the point of view of small satellites and CubeSats. Accordingly, MIMPS-G is not a "single-purpose" or "one-time-use" propulsions system, otherwise, it is designed to fulfill different space missions with various $\Delta V$ requirements relying on the modularity and expandability properties, where the 1U main propulsion module is capable of delivering at least total impulse of $I_{tot}$ = 850 N s with the possibility to add extension tanks of at least $I_{tot}$ = 1100 N s per tank, theoretical values. The baseline design of the 1U main propulsion module relied on studying orbital maneuver requirements of different CubeSat missions; examples are presented in the following paragraph.

Modern CubeSat missions have evolved from technology demonstration missions to real missions involving long-life commercial applications and scientific space exploration. Big economies are growing around "Earth Observation Services" as an example, that are mainly provided by private sector players, thanks to the small satellites industry, particularly CubeSats. Such commercial missions that rely on operating small satellite

constellations in significantly low earth orbits (LEO) require a dedicated propulsion system onboard the spacecraft to ensure long life and maximum profitability. These types of missions and applications require active orbital operations such as formation flying, attitude control, and drag compensation, especially in orbits subject to rigorous atmospheric drag. Recently, due to the growing number of satellite constellations, obstacle avoidance maneuvers in crowded LEO orbits impose high-thrust impulsive capabilities. Table 1 shows $\Delta V$ requirements for drag compensation and lifetime extension of nanosatellites in LEO. Tables 1 and 2 present data derived by Nardini, F. T. et al. [10]. Table 1 considered that the 1U and 3U spacecraft are of 1 and 4 kg, respectively, while the 8 and 10 kg spacecraft are of 6U standard size, all with the small cross-section facing the flight path; Data were derived using the NRLMSISE-00 atmospheric model, assuming a drag coefficient $C_D = 2.2$ and no deployable panels for standard CubeSat sizes. As for scientific deep-space exploration demonstrated in Lunar and interplanetary missions, orbital transfers require a significant $\Delta V$ budget. Table 2 presents different orbital transfer maneuvers and the required $\Delta V$ utilizing relatively high-thrust impulsive shot maneuvers. Clear assumptions were not mentioned or explained by the source [10] regarding the derivation of some data in Tables 1 and 2, such as the precise method of calculation for the lifetime and the burn duration in case of impulsive shot maneuvers; the values of $\Delta V$ for LEO to GEO and LEO to Lunar Orbit transfers are quite similar and clear calculations are not explained, therefore these data were taken as generic reference and were not applied in any calculations during the design phase of our propulsion system.

**Table 1.** Drag compensation for nanosatellites in LEO [10].

| Orbit Altitude (km) | Spacecraft Mass (kg) | Lifetime (y m d) | $\Delta V$ for 50% Increase Life-Time (m s$^{-1}$) |
|:---:|:---:|:---:|:---:|
| | 1 | 1.3 d | 9.28 |
| 200 | 4 | 4.4 d | 7.92 |
| | 8 | 2.8 d | 8.80 |
| | 10 | 3 d | 8.57 |
| | 1 | 21.8 d | 11.96 |
| 300 | 4 | 2 m 26 d | 11.67 |
| | 8 | 1 m 22 d | 11.77 |
| | 10 | 1 m 26 d | 15.76 |
| | 1 | 6 m 13 d | 14.20 |
| 400 | 4 | 2 y 1 m 11 d | 13.77 |
| | 8 | 1 y 3 m 12 d | 14.01 |
| | 10 | 1 y 4 m 18 d | 14.01 |

**Table 2.** Orbital changes $\Delta V$ using impulsive shot maneuvers [10,11].

| Maneuvers | $\Delta V$ (km s$^{-1}$) |
|:---:|:---:|
| LEO to GEO [a] | 3.95 (no plane change) |
| GTO to GEO | 1.5 (no plane change) |
| LEO to Earth Escape | 3.2 * |
| LEO to Lunar Orbit | 3.9 |
| GTO to Lunar Orbit | 1.7 |

[a] Calculated using Edelbaum's equation. * For jet exhaust to initial circular velocity ratio = 10.

## 1.2. Green Monopropellants Trade-Off Study

ASCENT or the Advanced SpaceCraft Energetic Non-Toxic propellant, formerly known as AF-M315E for Air Force Monopropellant, was developed by the Air Force Research Laboratory AFRL in 1998 [12]. This propellant is a hydroxylammonium nitrate HAN-based green monopropellant, and when decomposed produces an adiabatic flame temperature of about 2100 K which is much higher than that of the classic monopropellant hydrazine (~1200 K). ASCENT offers a 63% increase in density and a 13% increase in specific impulse over hydrazine [13], which makes it better in the miniaturization of

propulsion systems over the latter. The theoretical vacuum specific impulse $I_{sp}$ ranges from 260 to 270 s depending on the evaluation conditions. This propellant possesses high solubility and negligible vapor pressure of all its solution constituents, thus promoting high mixture stability at a wide range of temperatures, and low toxicity hazards in development and testing environments [14]. The favorable solubility and vapor pressure properties were found to be interesting, particularly for the micro electric pump feed system development. An advantage ASCENT possesses over most current state-of-the-art green propellants is its maturity. Thorough development of HAN-based propellants has taken place since the beginning of the development program of the Liquid Gun Propellants (LGP) by the U.S. Army until reaching this product and was tested in space on 1 N and 22 N thrusters through the GPIM Green propellant Infusion Mission launched in 2019 [15].

LMP-103S is the most mature among the ammonium dinitramide ADN-based green propellants and was qualified by ESA the European Space Agency and was in-space demonstrated through the High-Performance Propulsion System (HPGP) on Mango-PRISMA satellite launched in June 2010 [16,17]. Advantages of LMP-103S over ASCENT include lower combustion temperature which allows using materials with lower melting point and simpler designs for the thruster development. The adiabatic flame temperature of LMP-103S is around 1900 K while its theoretical vacuum specific impulse $I_{sp}$ is about 250 s. FLP-103, 105, 106, and 107 are other examples of ADN-based propellants that were developed by the Swedish Defense Research Agency (FOI) in Europe in 1997 [18–20]. FLP-family of propellants possess thermophysical properties close to LMP-103S and their performance and composition are highlighted in Table 3. In addition, ADN-based green monopropellants showed flexibility in using different ignition techniques other than catalytic decomposition, as demonstrated in lab experiments [16,21]; this may allow for the development of novel monopropellant thruster designs.

**Table 3.** ADN-based monopropellants properties [18,22,23] (ideal vacuum $I_{sp}$ by [22] using NASA CEA @ 2.0 MPa chamber pressure, 50:1 expansion ratio assuming frozen condition [16]).

| Propellant | Formulation | $I_{sp}$ (s) | $\rho$ (g cm$^{-3}$) | $\rho I_{sp}$ (g s cm$^{-3}$) | $T_c$ (K) |
|---|---|---|---|---|---|
| LMP-103S | [1] 63.0% [2] 18.4% [6] 18.6% | 252 | 1.24 | 312.48 | 1903 |
| FLP-103 | [1] 63.4% [2] 11.2% [5] 25.4% | 254 | 1.31 | 332.74 | 2033 |
| FLP-106 | [1] 64.6% [3] 11.5% [5] 23.9% | 255 | 1.357 | 344.6 | 2087 |
| FLP-107 | [1] 65.4% [4] 9.3% [5] 25.3% | 258 | 1.351 | 348.5 | 2142 |

[1] ADN. [2] Methanol. [3] MMF. [4] DMF. [5] Water. [6] Ammonia (aq. 25% concentration). @ 20 °C.

HNP (Highly stable Non detonating Propellant) is a HAN/HN-based family of green monopropellants developed by IHI Aerospace of Japan. This family includes HNP209, HNP221, and HNP225, and they are formulated from hydroxyl ammonium nitrate (HAN), hydrazinium nitrate (HN), methanol, and water [24]. HNP225 is the one among the family with the least adiabatic flame temperature, approximately 1000 K, even less than hydrazine (~1200 K), and delivers theoretical vacuum specific impulse $I_{sp}$ almost 200 s [24,25], properties shown in Table 4. The low-temperature combustion gasses of HNP225 allowed for the development of low-cost 3D printed thrusters since the requirement for high heat resistant materials for the thruster's combustion chamber is no longer present [26]. The HNP family of green monopropellants ignite using catalytic decomposition. Igarashi et al. 2017 [25] performed tests with newly developed proprietary catalysts and showed excellent response and stability compared to hydrazine, either in pulsed mode operation or continuous mode, with preheating temperatures starting from 200 °C for HNP221 and HNP225 monopropellants. HNP2xx family performance chart represented in Figure 2 provides for comparison with hydrazine and state-of-the-art EILs, as well as highlighting the melting point of Inconel® 625.

**Table 4.** Performance and physical properties of HNP2xx green monopropellants family [25] as cited in [27].

| Propellant | Theoretical Vacuum $I_{sp}$ (s) | Density $\rho$ (g cm$^{-3}$) | Volumetric $\rho I_{sp}$ (g s cm$^{-3}$) | Chamber Temp. $T_c$ (K) |
|---|---|---|---|---|
| HNP209 | 260 | 1.32 | 343 | ~1900 |
| HNP221 | 241 | 1.22 | 294 | 1394 |
| HNP225 | 213 | 1.16 | 245 | 990 |

@ 1.0 MPa chamber pressure, 100:1 expansion ratio, and ideal vacuum conditions.

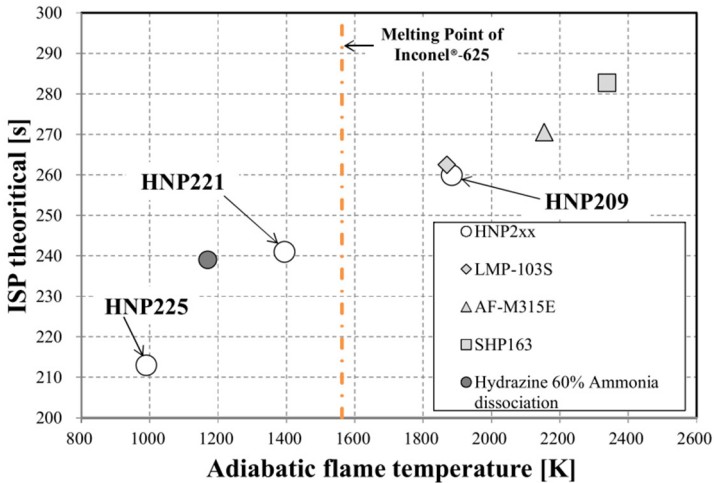

**Figure 2.** HNP2xx green monopropellants family performance chart compared to other EILs. Adapted from Igarashi and Matsuura 2017 [25] with permission.

Among the state-of-the-art green monopropellants surveyed above, four EILs were considered for a trade-off study, Table 5, either for their maturity or for their promising potential. During the process of nominating candidate propellants for the propulsion system design, the rocket performance characteristics of each propellant (such as the volumetric specific impulse) were not the main focus as the selection criteria. Significant attention was put on the propellants' thermochemical characteristics (i.e., the adiabatic flame temperature) since the lower adiabatic flame temperature would impact the thruster design simplicity as well as mass and costs reduction. The rest of the selection aspects such as operation pressure-temperature conditions, service temperature and vapor pressure were placed according to the typical requirements of the spacecraft propulsion systems under study. The characteristics of the proposed EIL green monopropellants for MIMPS-G500mN propellant trade-off study and the propellants trade-off requirements are presented in Tables 5 and 6, respectively.

**Table 5.** Performance and physical properties of the proposed EIL green monopropellants for MIMPS-G500mN [14,16,21,25] as cited in [27].

| Propellant | Theoretical Vacuum $I_{sp}$ (s) | Density $\rho$ (g cm$^{-3}$) | Volumetric $\rho I_{sp}$ (g s cm$^{-3}$) | Chamber Temp. $T_c$ (K) | Freezing Temp. $T_F$ (°C) | Vapor Pressure (kPa) | Maturity |
|---|---|---|---|---|---|---|---|
| AF-M315E | 266 | 1.47 | 391 | 2166 | <−80 | 1.4 | High |
| LMP-103S | 252 | 1.24 | 312.48 | 1903 | −7 | 13.6 | High |
| FLP-106 | 255 | 1.357 | 344.6 | 2087 | 0 | 2.1 | Medium |
| HNP225 * | 213 | 1.16 | 245 | 990 | ≤−10 | uncertain | Low |

Evaluation conditions 2.0 MPa chamber pressure and $A_e/A_t$ 50:1; * 1.0 MPa and $A_e/A_t$ 100:1. Vapor pressure at 25 °C.

**Table 6.** Propellants' trade-off study requirements.

| Requirement | Description |
|:---:|:---:|
| 1 | Use of Green propellant complying with ECHA–REACH directive articles. |
| 2 | Use of monopropellants classified as EIL. |
| 3 | EIL Green Monopropellants should have specific impulse performance of $I_{sp} \geq 200$ s. |
| 4 | Freezing temperature of the propellant shall be $\leq -10\ °$C. |
| 5 | Propellant must be liquid within pressure range [0.1, 3] MPa and temperature range $[-30, +80]\ °$C. |
| 6 | Propellant shall possess Low Vapor Pressure, typically below 20 kPa at room temperature (LMP-103S is ~14 kPa @ 25 °C [16]). |

Trade-off criteria in Table 7 were set to fulfill previously elaborated design goals and the rationale behind each criterion is described in the following. The first criterion is the specific impulse $I_{sp}$ (s) which is by definition one of the most important performance parameters in the design and evaluation stages. Generally, $I_{sp}$ increases with higher combustion temperatures and by burning propellants possessing lighter and molecularly simpler combustion products. As highlighted before, the higher the value of such parameter is not necessarily the better for the system performance. Thus, an optimal value must be chosen to achieve considerable overall performance while maintaining suitable system inert mass and components' design simplicity and cost; this is possible when considering the thruster's material choice which is highly coupled with the resulted adiabatic flame temperature. The $I_{sp}$ (s) criterion was evaluated for the considered propellants by a knock-out condition, that the considered propellants shall possess $I_{sp} \geq 200$ s as expressed in requirement three in Table 6, all propellants fulfilling this criterion shall score equally the highest score. The second criterion is the volumetric specific impulse $\rho I_{sp}$ (g s cm$^{-3}$), generally, the higher propellant density shall occupy lower tank volume, thus a higher value is considered better, and the score is evaluated accordingly. The third criterion, the decomposition chamber temperature $T_c$ (K), is one of the most important parameters in this trade-off study, as conceptualized earlier. The lowest decomposition chamber temperature value is considered the best among all considered propellants, and a weight factor of ($\times 2$) is imposed to emphasize the importance of this criterion. Freezing temperature $T_F$ (°C), or service temperature as more accurately described, since some EILs undergo precipitation [28] or glass transition as in the case of AF-M315E [29], is the fourth criterion assessed in the trade-off study. A low freezing point is required for the propellant's storable and operational stability over a long time and is important to reduce tank heating power consumption. The last criterion is the vapor pressure $P_{vap}$; EIL green monopropellants are characterized by very low vapor pressure that allows for stable ground testing, storability, and transportability as well as in-space operability. Since this study focused on unconventional autogenous pressurization, the use of low vapor pressure propellants is crucial for the propulsion system's operational stability. Higher vapor pressures, to some extent, would definitely optimize the use of electric heating power for thruster feed and tank pressurization, however, in early development phases the lower vapor pressure is more appreciated. Table 7 presents the propellant trade-off criteria, methods of calculation and evaluation for each criterion as well as the value function considered.

The "Value Function" is a tool to assist in scoring each propellant against the trade-off criteria. Two main types of value functions are used, namely "The Higher the Better" and "The Lower the Better", and another one is a knockout condition. The latter condition would discard any propellant with theoretical specific impulse <200 s, while the other two value functions will be graded on a 0–10 scale, with the minimum and the maximum values depending on each value function type, refer to Figure 3.

**Table 7.** Propellant trade-off criteria.

| Trade-off Criteria | Symbol | Method of Calculation | Value Function |
|---|---|---|---|
| Specific Impulse | $I_{sp}$ (s) | RPA simulations and literature. | Knockout condition per Requirement #3 |
| Volumetric Specific Impulse | $\rho I_{sp}$ (g s cm$^{-3}$) | RPA simulation and Propellant Thermodynamic properties Literature. | The higher the better |
| Combustion Temperature | $T_c$ (K) | RPA simulation and Propellant Thermochemical Literature. | The lower the better |
| Freezing Temperature | $T_F$ (°C) | Literature | The lower the better |
| Vapor Pressure | $P_{vap}$ (kPa) | Literature | The lower the better |

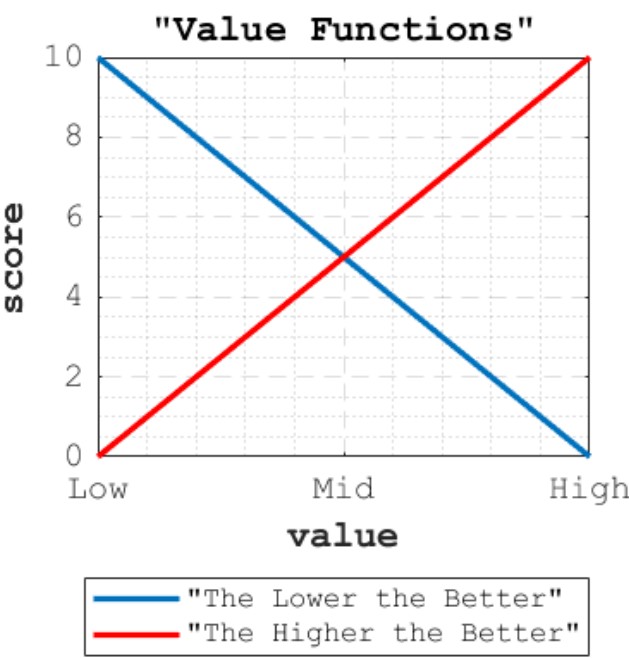

**Figure 3.** Value Function scoring graph.

Propellant characteristics and performance parameters for the four propellants considered in the trade-off study (i.e., AF-M315E–LMP-103S–FLP-106–HNP225) are presented in Table 5. The values of performance parameters and propellant thermochemical properties were evaluated for some propellants using Rocket Propulsion Analysis (RPA) analysis tool academic version, along with a literature review for other propellants with proprietary formulations. The physical and thermodynamic properties of all propellants were collected from the literature. Finally, the propellant trade-off results and ranks are presented in Table 8.

**Table 8.** Trade-off study results and propellants ranking.

| Propellant | Score per Criterion | | | | | Overall Score (Ranked) |
|---|---|---|---|---|---|---|
| | $I_{sp}$ | $\rho I_{sp}$ | $T_c$ | $T_F$ | Vapor Pressure | |
| AF-M315E | 10 | 10 | 2 | 10 | 10 | 42 |
| HNP225 | 10 | 0 | 20 | 2 | 0/Uncertain | 32 |
| LMP-103S | 10 | 5 | 6 | 3 | 3.5 | 27.5 |
| FLP-106 | 10 | 7 | 0 | 0 | 9 | 26 |

### 1.3. Unconventional Feed and Pressurization Systems

Autogenous pressurization is an old concept that has been utilized in space systems since 1968 [30] and it is mainly used in medium to large size pump-fed engines. The system uses vaporized propellants to pressurize tanks by passing streams of cool propellant through a heat source that can be the thrust chamber cooling jackets or heat exchangers. This term was sometimes paired with turbopump feed cycles, especially in launcher engines. Nowadays, electric pump feed cycles are a major focal point in various current research work, especially after being successfully utilized in the Rutherford engines of the Electron launch vehicle developed by Rocket Lab [31] and started to be more frequently proposed nowadays for small and medium rocket engines.

Micro electric pump feed cycle can be considered unconventional from the perspective of micro and nanosatellite development. In such a system, low ullage pressure is maintained in a way to provide propellant to the pump at required pump suction conditions, which is essential for stable feed operation and to protect against pump cavitation and pressure pulsation. Low propellant storage pressure levels are needed for pump-fed systems in general, typically 0.07 to 0.34 MPa [32–34] and these values may slightly increase in the case of CubeSat small tanks. Much lighter tank structures are used in the case of pump-fed systems due to this required low storage pressure but they still come at the cost of high system complexity of the pump operation and the accommodated propellant feed lines. Although pump-fed systems are not widely used in the current time for CubeSats, possibly only proposed, the technological advancements in micro electric pumps development, such as the relatively low-cost micro electric pump [35] used in the proposed design, show the possibility to use this feed and pressurization technique on the scale of micro and nano spacecraft.

Autogenous-pressurization will impose challenges especially considering the nature of EIL salt solutions. The observed fact of slow decomposition of some propellant mixtures, such as the ADN-based LMP-103S, leading to salt residues and solid particles precipitation in thrusters' valves impose design challenges and limitations that need to be carefully addressed. In the case of vaporization of propellant streams, the risks relating to operation stability are amplified due to the expected precipitation of salts, especially in the small feed tubing, microvalves, and pump. A range of solutions was proposed during the preliminary design and analysis phases, that are of a mechanical nature, such as adding proper filtration, incorporating synchronous flushing procedures, or more dedicated and accurate sizing for the tubes and piping in more sensitive areas to avoid such problems. EIL propellants with significantly high solubility and non-detonable nature would be highly appreciated, which was one of the reasons HNP225 was chosen for the development phases. However, more insights will be gained during the development and testing phases and more design iterations are expected to be carried out in the feed and pressurization system to counter development challenges.

## 2. System Analysis and Design Methodology

MIMPS-G500mN is designed to suit space missions demanding high-thrust impulsive orbital maneuvers and to be able to perform various active orbital operations. Moreover, the system should provide design modularity and expandability in terms of propellant capacity and thrust levels and in order to accommodate high total impulse and maneuverability requirements of long duration and interplanetary space missions. Accordingly, technical requirements were defined while accounting for different manufacturing and development considerations. Figures 4 and 5 show the unified modeling language UML schematic diagram of the design process, and the preliminary design flow chart, respectively, and all will be discussed in the following subsections.

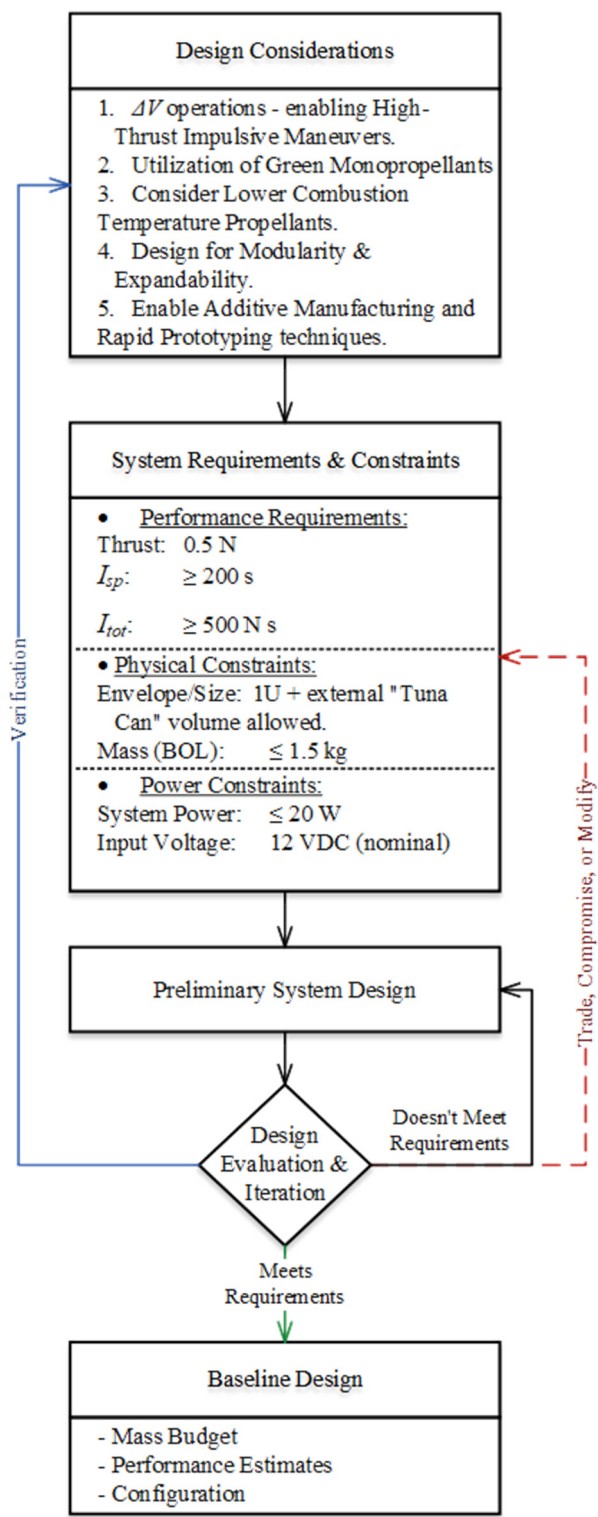

**Figure 4.** UML schematic diagram of MIMPS-G design process.

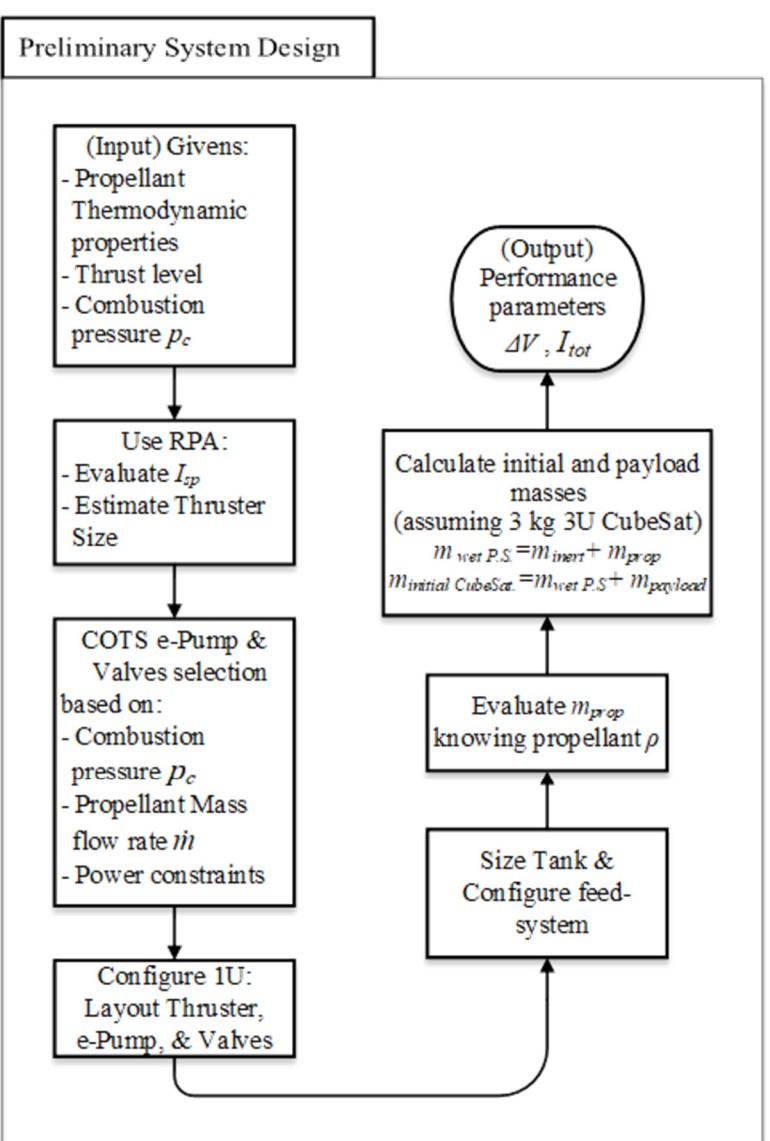

**Figure 5.** Preliminary design flow chart.

## 2.1. Requirements and Design Considerations

The propulsion system designated MIMPS-G is aimed to be a primary propulsion system that enables high-thrust impulsive maneuvers. MIMPS-G operates on green monopropellants classified as Energetic Ionic Liquids (EILs). The research interest and focus were oriented toward studying HAN-based and ADN-based propellants, and a special focus was given to low decomposition chamber temperature formulations. As ASCENT (formerly AF-M315E) is considered one of the most mature and widely used and proposed green monopropellants, other alternatives were sought to allow variation in propellant sources and performance improvement. During the mechanical design of the propulsion system, emphasis was made on system modularity and expandability, where the former will allow to easily orient components within a spacecraft with different standard CubeSat sizes and make the best use of allowable space. The latter, namely the expandability, is a unique design criterion that will further impact CubeSat utilization and clustering of COTS propulsion systems. Along with modularity, the expandability property will give the ability to increase propellant tanks and even thrust levels on a "plug-and-play" basis. Recently, researchers in the field of small satellites are seeking rapid prototyping and low-cost manufacturability [25] by employing additive manufacturing techniques. Metal 3D printing nowadays utilizes exotic space materials such as Inconel-625® and Ti-6Al-4V

(Ti64). This manufacturing technique will help in reducing the parts number in a given design and thus overall part mass, as well as reducing prototyping and manufacturing processes lead-time.

Design requirements, refer to Figure 4, imply having a thrust level of 0.5 N, gravimetric specific impulse $\geq$200 s, and total impulse $\geq$500 N s which is almost the lowest value available in the market in this class of commercial propulsion systems. Choosing an 0.5 N thruster was the maximum possible value from a single thruster to be integrated into a 1U unit size due to size and dimensions constraints. Moreover, when comparing 0.1 to 0.5 N thrusters, the higher value is considered better when employing impulsive maneuvers which translates to lower burn time. In addition, the high thrust becomes the dominant term between the external forces acting on the satellite such as gravitational forces and significant drag forces in very low orbits, (Section 9.1.1 in [36]); all these reasons contribute to better efficiency of high-thrust propulsion systems. Another point that favors the 0.5 N thruster is "clustering"; in the case of using a lightweight 3U satellite, 0.1 and 0.5 N thrust levels would not have a significant impact on performance, but in the case of clustering several propulsion modules to a larger size spacecraft (such as Figures A4–A6 (Appendix A)), every available newton of thrust will contribute significantly to the maneuver efficiency. Regarding the specific impulse value to be $\geq$200 s, two points have introduced this value, first one is the specific impulse value of hydrazine (i.e., ~236 s theoretical vacuum) and the other is the value associated with high concentration hydrogen peroxide (e.g., HTP 98% $\approx$ 186 s); the previous implies that the value of the used green propellant should be at least 200 s to outperform hydrogen peroxide as green monopropellant and still maintain a relatively high performance if considering the classic toxic hydrazine. Concerning the value of the gravimetric specific impulse mentioned, as widely interpreted in design literature, the higher $I_{sp}$ is considered better, but this is not always the correct interpretation since it mostly comes usually at cost of higher combustion temperatures, and thus higher weight materials used in thruster's development and thermal management. Of course, $I_{sp}$ depends on both combustion temperature and molecular mass of a given propellant, and high $I_{sp}$ can still be acquired at relatively low temperatures if the molecular mass of decomposition products is lighter and molecularly simpler. Therefore, choosing an optimal specific impulse value, not necessarily a high value, for a given propellant that tends to have lower adiabatic flame temperature will impact positively on the propulsion system's overall performance, cost, and project lead-time. A thruster with low-weight materials might not necessarily have a great impact on the propulsion system mass reduction, however, in the case of enabling Additive Manufacturing (AM) techniques, a further limitation on combustion temperature is imposed to respect the melting point of certain 3D printing materials such as Inconel-625® (~1563.15 K). In the study phase presented in this article a commercial thruster model operating on high combustion temperatures was considered for the preliminary design. Further project phases will consider the design and development of metal 3D-printed thrusters that operate only low adiabatic flame temperature monopropellants. The physical constraints set on the design imply maintaining a standard CubeSat size of 1U while considering the extra protrusion for the thruster referred to as the "Tuna Can" volume. The size of this extra volume occupies the ejection spring of the CubeSat deployer and varies from one model to another and depends on the manufacturer [37–39]. A suitable deployer allowing a protrusion volume of φ86.0–78.0 mm was considered. The initial Beginning Of Life (BOL) mass requirement set was $\leq$1.5 kg for 1U in order to have a competitive advantage over state-of-the-art commercial propulsion systems; it will be shown in the following design sections that this requirement was partially fulfilled since reducing the propellant mass for the denser propellants was required to maintain this value (i.e., $\leq$1.5 kg). Otherwise, the requirement can be modified by increasing the constraint to get the use of allowable propellant volume in the tank. As for the electric power requirements, a system power of $\leq$20 W and nominal 12 VDC was considered after studying the electrical properties of the various system parts and such details will be briefly highlighted in the mass budget table.

### 2.2. Design Process and Methodology

The preliminary design process did not follow the conventional design flow of rocket propulsion that usually starts by identifying a specific mission and assess its $\Delta V$ requirements and further proceeding with a design to fulfill this requirement, perhaps among others, for this unique mission. However, in the used approach, broad types of space missions were surveyed to highlight orbital maneuvers requirements and to set a baseline for the $\Delta V$, total impulse, and thrust level requirements. As for longer duration or interplanetary missions, clustering of the propulsion system with simultaneous operation of different parts, refer to Figures A4–A6 (Appendix A) will be the main player in further extending the $\Delta V$, total impulse, and thrust level requirements beyond the baseline. From this point, and referring to the previously mentioned design considerations, the design flow proceeded with identifying and allowing for a maximum allowable propellant volume for a 1U standard unit size. The development and use of the unconventional and novel, with respect to CubeSats, autogenous pressurization and a micro electric pump feed system concept was the main aspect behind reaching a new maximum allowable propellant volume as compared to conventional pressure-fed systems; refer to Figure 5 for the preliminary design flow chart.

### 2.3. MIMPS-G ConOps

Micro e-Pump feed system is considered unconventional for in-space propulsion, especially for small-sized spacecraft. The electric pump feed system (see Figures 6, A2 and A3) is primarily responsible for the delivery of propellant from very low storage pressure to high-pressure requirements of the thrust chamber at a given propellant mass flow rate ($\dot{m}$). Moreover, it is required to circulate streams of propellant over a heat source for vaporization and to use the vapor (non-catalytically decomposed) of the liquid propellant to keep the storage tank at the required minimal pressure levels for proper pump operation—typically describes an autogenous pressurization system. The heating of the propellant streams will take place through radiative heat transfer from the decomposition chamber. A preliminary clearance value was set between the thruster chamber wall and the spiral tubing intended for propellant stream heating; the reason behind this was, first, to avoid unneeded power consumption due to heat loss to the propellant stream tubes during the pre-heating phase of the thruster's catalyst, and secondly, to eliminate any risk of transient heat loss on introducing propellant streams during the operation phase. Accurate sizing of the spiral tubing is pending verification, considering heat transfer estimation and propellant vaporization characteristics. The design of this part is expected to undergo several iterations and modifications in the prototyping and testing phases. One of the advantages of this concept is that no separation within the tank is required–no need to separate the feed-back vapor unlike the case of feeding back catalytically decomposed gaseous propellant–thus avoiding actuating and separation mechanisms as in case of piston expelled tanks, or material compatibility problems with green propellants as in the case of using bellows or elastic diaphragms.

An optional vapor auxiliary propulsion for reaction control and attitude control requirements can be integrated. This optional subsystem incorporates a small catalytic bed and lighter weight thrusters compared to the primary monopropellant thruster and shall present a "multimode" propulsion system when incorporated—multimode propulsion is capable of utilizing the same propellant tank for different types of propulsion at the same time [6,40,41]. The catalytic bed shall increase the temperature of the vapor, thus increasing performance, moreover, ensures homogenous exhaust. This concept is complemented and reinforced by the research work of Rhodes and Ronney (2019) on the $H_2O_2$ vapor propulsion system [42]. Of course, HAN and ADN-based propellants differ completely from $H_2O_2$, but the analogy intended here is in using the propellant vapor over the relevant catalytic bed to increase its temperature and ensure homogeneity, thus increasing the propulsion performance. The vapor auxiliary system modeling will not be incorporated in the design stage presented in this article.

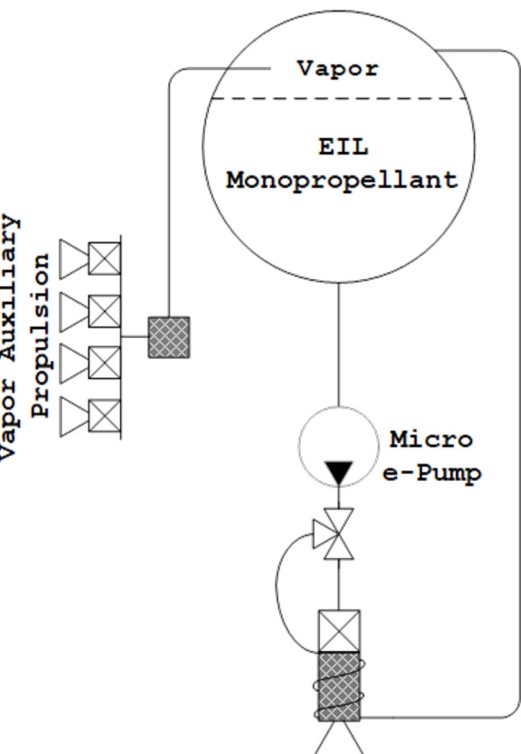

**Figure 6.** MIMPS-G propulsion system schematic diagram (including the optional auxiliary vapor propulsion in multimode architecture).

A micro three-way flow control valve is required to control the flow from the micro e-Pump outlet to the thruster and the propellant streams heating cycle. It should be observed that technological advancements in, and availability of, controlled microvalves are inevitable for such feed and pressurization system to succeed on this small-size scale. One final advantage to mention, but not the last, for this feed system is the ability to precisely control the propellant mass flow rate ($\dot{m}$) and pressure ($p_c$) to the thruster chamber, thus controlling and maintaining a constant thrust level over almost the whole mission life-time. This unconventional concept may also be applicable for feed and pressurization systems of liquid bipropellant propulsion of small-satellites and spacecraft.

### 3. Preliminary Design Study

As presented in the flow chart in Figure 5, the preliminary design of the MIMPS-G500mN propulsion system started by assessing the thermodynamic, thermochemical, and performance characteristics of the selected EIL green monopropellants. Rocket Propulsion Analysis (RPA) academic version was used in propellants assessment for propellants with precise known formulation, such as LMP-103s and FLP-106, the inputs for the analysis tool were the chemical formulae of constituents, molecular weights, standard heat of formation, and weight percent of the formula constituents. Predefined values given to the analysis tool for the monopropellant engine were 500 mN thrust value, moreover, iterations of simulations were made between 1–2 MPa combustion pressures. The nozzle expansion ratio was also iterated between 50 to 100:1. Further, thermodynamic properties of the simulated propulsion system were extracted, such as the chamber temperature, specific heats, and specific heat ratio at the thruster different regions. Theoretical (ideal) performance as well as estimated delivered performance were assessed, namely the effective exhaust velocity and the weight-specific impulse at vacuum condition. Other proprietary propellants such as ASCENT (formerly AF-M315E) and HNP225, with unknown precise formulation weight fractions, were not possible to be simulated in the analysis tool, thus it was relied on the published literature by propulsion system developers and manufacturers to acquire the above-mentioned data.

Micro electric pump and microvalves were chosen COTS parts based on the operation pressures, propellant mass flow rate, size constraints, and electric power constraints. As mentioned before, the thruster considered in the preliminary design is a commercial model by Busek company that is the 0.5 N green propellant thruster [43–45]. After laying out the main propulsion system components, see Figure A2 (Appendix A), the propellant tank was structurally sized and verified for operation pressures, temperatures, material compatibility, and design modularity and expandability. The tank will use a Propellant Management Device (PMD) consisting of vanes and a sponge on the outlet with light-weight compatible materials to the green monopropellants. The structural design of the tank considered a titanium wetted inner structure reinforced by carbon fiber composites on the outside to ensure long-term propellant material compatibility [46]. The tank design dedicated a rough 10% and 5% of the volume for the PMD and ullage, respectively.

### 3.1. Equations and Formulae

The following are the fundamental equations of ideal rocket theory that are used to produce the design data.

$$I_{tot} = I_{sp} \, m_{prop} \, \mathbf{g}_o \tag{1}$$

$$m_f = m_i - m_{prop} \tag{2}$$

$$m_i = m_{wet \ P.S.} + m_{payload} \tag{3}$$

$$m_{wet \ P.S.} = m_{inert} + m_{prop} \tag{4}$$

$$\Delta V = -I_{sp} \, \mathbf{g}_o \, \ln\left(\frac{m_f}{m_i}\right) \tag{5}$$

$m_{wet \ P.S.}$: is the wet mass of the propulsion system.
$m_{prop}$: is the propellant mass.
$m_{inert}$: is the inert or dry mass of the propulsion system.
$m_i$: is the initial mass of the propulsion system.
$m_{payload}$: the payload here is considered any and every part outside the propulsion system envelope (not only the payload of the spacecraft).
$m_f$: is the final mass of the propulsion system.

## 4. Results and Discussion

The main propulsion module storage tank empty volume is 420 cm$^3$ and after considering the PMD and ullage volume of 15% of this value, the allowable propellant volume becomes 357 cm$^3$, refer to Figure 7. The allowable propellant volume for the extension tank is 474.16 cm$^3$, Figures A4–A6 (Appendix A), considering 20% PMD and ullage. Furthermore, the mass of each propellant along with the total impulse is calculated and presented in Table 9 using the fundamental equations of ideal rocket theory explained in Section 3.1.

**Table 9.** MIMPS-G total impulse $I_{tot}$ with the selected green monopropellants.

| Total Tank Empty Volume = 420 cm$^3$ PMD and Ullage = 15% Allowable Propellant Volume = 357 cm$^3$ | | | |
|---|---|---|---|
| | **Propellants** | | |
| | **AF-M315E** | **HNP225** | **LMP-103S** |
| $\rho$ (g cm$^{-3}$) | 1.4699 | 1.15023 | 1.2420 |
| $m_{prop}$ (g) | 524.75 | 410.632 | 443.394 |
| $I_{tot}$ (N s) | 1369.310 | 858.027 | 1096.123 |
| **Extension Tank Allowable Prop. Volume = 474.1 cm$^3$** | | | |
| $I_{tot}$ (N s) | 1818.721 | 1139.627 | 1455.859 |

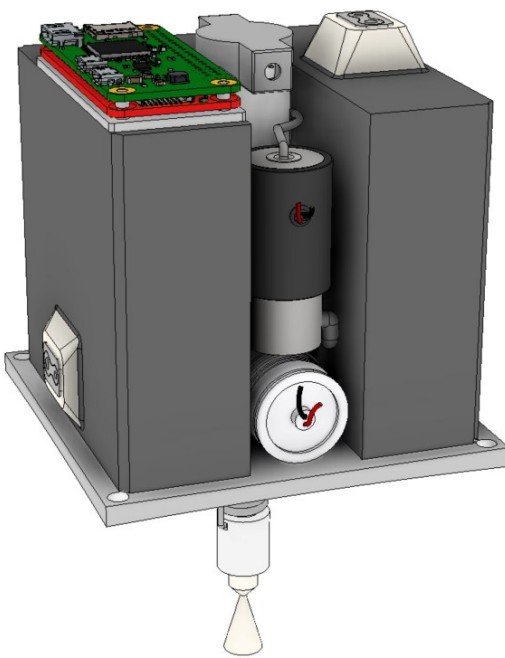

**Figure 7.** MIMPS-G500mN main module showing the tank shape and different system components.

The propulsion system mass budget of dry components is presented in Table 10. COTS components data were collected from the data sheets according to the design requirements. The propellant storage tank was sized to operate at approximately 0.7 MPa nominal pressure, and a 1.2 MPa MEOP was considered, moreover, a pressure burst of 2.0 MPa was calculated by imposing approximately 2.8 factor of safety. Furthermore, a design margin was considered for manufacturability, thus having at least 1 mm thickness titanium wetted part and an outer carbon fiber composite reinforcement plies of 2 mm thickness. The tank also accommodates polyimide Thermofoil™ heaters, a pressure sensor, and a temperature sensor, while a mass flow sensor of Out-of-Liquid type should be attached to the tank outlet pipe at least in the testing phase. The micro three-way flow control valve is made of the state-of-the-art acetal homopolymer Delrin® that possesses great anti-corrosion properties and is lightweight. The PMD consists of a combination of vanes and a sponge structure at the outlet considering Delrin® and Titanium alloy for these parts, respectively. The mass of the storage tank and the feed system was calculated using the CAD modeling software while considering a conservative error margin.

**Table 10.** Inert mass budget of MIMPS-G500mN propulsion system.

| Part | Materials/Comments | Mass (g) |
|---|---|---|
| Cover | Carbon Fiber Reinforce Composites $\rho$ = 1.430 g cm$^{-3}$ | 65 |
| Base | Aluminum 6061-AHC $\rho$ = 2.79 g cm$^{-3}$ Carbon–Carbon Laminate $\rho$ = 1.7 g cm$^{-3}$ | 101 |
| Micro e-Pump | COTS micro gear pump (7 W–12 VDC) | 75 |
| 3-way solenoid micro FCV | COTS Acetal polymer (Delrin®) [a] Material Compatibility A-Excellent with Alcohols and aqueous Ammonium nitrate [46–48] (2 W) | 45 |
| Piezo Microvalve–Thruster FCV | Piezo tech/Titanium-wet (200 mW) | 67 |
| Thruster 0.5 N | Niobium/Titanium (Heaters 7–12 W; 12 VDC) without FCV | 80 |
| Storage Tank | CFRP 2 mm thick. $\rho$ = 1.430 g cm$^{-3}$ | 148 |
| | Ti64 1 mm thick. $\rho$ = 4.43 g cm$^{-3}$ | 228 |
| Tank I/O ports | 5 ports × 20 g *"Rough estimate"* | ~100 |

| Part | Materials/Comments | Mass (g) |
|---|---|---|
| Tank Heater | Polyimide Thermofoil$^{TM}$ Heaters (4 W; 6–12 VDC) | 4 |
| PMD [§] | Titanium alloys and Acetal (Delrin$^®$) Sponge and Vanes [49,50] (no steel, no CFRP) *"Rough estimate"* | ~50 |
| Microtube/Piping | Titanium alloy Grade 1 $\phi_{in}$ = 3 mm; t = 0.5 mm; total length = 363.6 mm | ≤10 |
| Computer, Sensors, and Interfaces | SBC *; Driver; 1 Pressure, 1 Temp. Sensors; Wiring | ≤120 |
| | Total Inert Mass | 1093 |

[a] Delrin$^®$ acetal homopolymer (Polyoxymethylene POM). [§] Propellant Management Device. * Single Board Computer. FCV: Flow Control Valve.

A control and computing unit was considered in the MIMPS-G design although the propulsion system control can be handled by the spacecraft main computer unit. The preliminary design considered extra free volume to allow for further tuning and tweaking of internal components. The current design is a result of many iterations to optimize available space, and components are placed to allow for dynamic stability of the spacecraft.

The physical properties and theoretical performance parameters of MIMPS-G500mN utilizing state-of-the-art green monopropellants are presented in Table 11. Although HNP225 has the lowest $I_{tot}$ and $\Delta V$, it allows for the greatest payload mass onboard the spacecraft while still complying with the design requirements and constraints mentioned in Figure 4 (i.e., ~1.5 kg BOL mass and $I_{tot}$ = 858.027 N s). If HNP225 is considered for MIMPS-G500mN it will allow for the use of metal 3D printed relatively low-cost thruster that would impact positively the propulsion system inert mass and thermal control due to the propellant low adiabatic flame temperature. The latter, along with the high solubility and non-detonating nature of this propellant, can be a point of advantage over other considered propellants in the first prototypes of the propulsion system with respect to management and control of the autogenous pressurization and feed cycle.

**Table 11.** Specifications and theoretical performance of MIMPS-G500mN using the selected green monopropellants.

| Propellant | AF-M315E | LMP-103S | HNP225 |
|---|---|---|---|
| Propulsion System Size | 1U + "Tuna Can" protrusion volume | | |
| $m_{inert}$ (g) | 1093 | | |
| $m_{prop}$ (g) | 524.75 | 443.394 | 410.632 |
| $m_{wet\ P.S.}$ (g) | 1617.75 | 1536.394 | 1503.632 |
| Spacecraft Size | 3 U–3 kg | | |
| $m_f$ (kg) | 2.47525 | 2.556606 | 2.589368 |
| $m_{payload}$ (kg) | 1.38225 | 1.463606 | 1.496368 |
| Thrust | 0.5 N | | |
| $I_{sp}$ (s) | 266 * | 252 * | 213 ** |
| $I_{tot}$ (N s) | 1369.310 | 1096.123 | 858.027 |
| $\Delta V$ (m s$^{-1}$) | 501.723 | 395.370 | 307.575 |

* @ 2.0 MPa chamber pressure and 50:1 expansion ratio. ** @ 1.0 MPa chamber pressure and 100:1 expansion ration [51].

A relatively low-cost COTS micro electric pump is used [35], that has a mass of only 75 g and cylindrical dimensions of ⌀22.0–70.60 mm, and provides propellant mass flow rate ($\dot{m}$) and output pressure up to 30 mL min$^{-1}$ and 2.2 MPa, respectively, at nominal 12 VDC and 7 W with viscous fluids similar to the used propellants, which makes this model a candidate for MIMPS-G. The "Tuna Can" protrusion volume existing within the CubeSat deployer springs differs from one deployer model to another which depends on the manufacturer. A deployer design allowing for protrusion volume of ⌀86.0–78.0 mm offered by a European manufacturer [39] was considered.

## 5. Market Survey on Small Satellites Monopropellant Propulsion

This section will discuss state-of-the-art small-sized spacecraft green monopropellant propulsion systems, refer to Figure 8. These propulsion modules are proposed for Earth-orbiting missions as well as Lunar missions. It was observed that propulsion systems manufacturers recently started to orientate toward unconventional feed and pressurization systems to overcome small size restrictions onboard small satellites while obtaining maximum total impulse performance possible. It was also seen that electric pump feed is being considered in very recently published and released work by Georgia Tech. and NASA for near future Lunar missions [52].

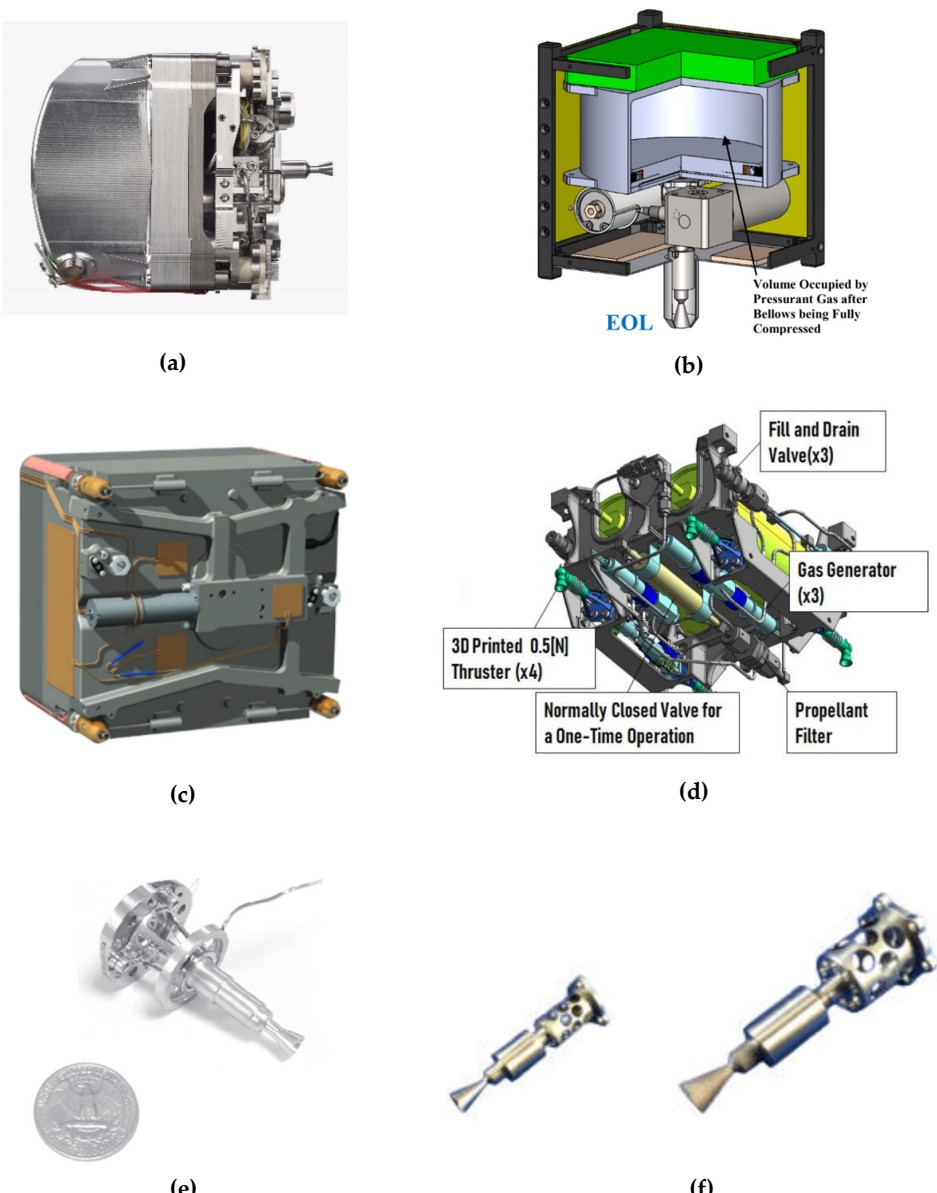

**Figure 8.** State of the art in green monopropellant propulsion: (**a**) EPSS C1 CubeSat Propulsion System. (Image courtesy of NanoAvionics [53]); (**b**) Busek 1U CubeSat Green Propulsion Module. (Image Courtesy of Tsay et al. [45]—Busek Co., Inc.); (**c**) MPS-135 Modular Propulsion System. (Image Courtesy of Aerojet Rocketdyne [54]); (**d**) Pinot-G propulsion module. (Image courtesy of IHI Aerospace Co. [55]); (**e**) HPGP 100 mN thruster. (Image Courtesy of Bradford-ECAPS); (**f**) BGT-X1 and BGT-X5 green monopropellant thrusters. (Image Courtesy of Busek Co.).

### 5.1. State-of-the-Art CubeSat Monopropellant Propulsion

State-of-the-art small satellites monopropellant propulsion were found to utilize thrusters with a range of thrust from 0.1 to 1 N, refer to Table 12. Some of the distinguished systems in the market are namely, EPSS C1 by NanoAvionics [53] in Figure 8a, BGT-X5 by Busek Company [12,43] in Figure 8b, and CubeSat Modular Propulsion System MPS-130 by Aerojet Rocketdyne [12,54] Figure 8c. The former system uses an ADN-based green propellant, while the latter two systems use the HAN-based ASCENT (formerly AF-M315E) [56]. Morris et al. [56] discussed the development of the MPS propulsion system with both hydrazine and the green monopropellant AF-M315E; the development and fabrication process used state-of-the-art additive manufacturing techniques using advanced space materials such as Inconel-625® and Ti-6Al-4V (Ti64) alloys which helped significantly in the development of these modular systems suitable for CubeSats envelope and mass constraints. All the above-mentioned systems utilize a pressurant gas either a conventional barrier separated pressure-fed system as in the case of EPSS C1 and MPS-130 or utilizing a post-launch gas generation mechanism such as the novel $CO_2$ gas generator in the case of BGT-X5 [45]. Nevertheless, new systems that are using pump feed with a propellant management device PMD are proposed by Aerojet Rocketdyne under the MPS propulsion system family, such as the MPS-135 4U and 6U [54]. Another recently market-released propulsion module series for small satellites is the Pinot-G developed by IHI Aerospace Japanese company [57]. Pinot-G burns HNP225 green monopropellant and incorporates four thrusters 0.5 N each. The wet mass of Pinot-G is 10 kg and occupies dimensions of ⌀350 mm diameter and 135 mm height. The pressurization technique relies on post-launch tank pressurization designed to be pressurized in orbit. The figures available on the company's website, refer to Figure 8d, show that three gas generators are connected to the propellant tanks, from both ends, which may be used in tank pressurization. A very interesting recent green monopropellant propulsion system for CubeSat is the one being designed and developed for the Lunar Flashlight Mission by the Georgia Institute of Technology and NASA [58]; this custom propulsion system is fueled by AF-M315E and fits in 2.5U standard size and maintains a total wet mass of less than 6 kg. This system relies on propellant pump feed and delivers over 2500 N s of total impulse.

**Table 12.** Performance data and specifications of state-of-the-art small satellites green monopropellant propulsion systems.

| Propulsion System | MPS-135 | BGT-X5 | EPSS C1 | Pinot-G | LFPS |
|---|---|---|---|---|---|
| Propellant | AF-M315E | AF-M315E | ADN-based blend | HNP225 | AF-M315E |
| Size | 4U | 1U | 1U | ⌀350–135 mm | 2.5U |
| Thrust (N) | 1 N × 4 thrusters | 0.5 N | 0.1 N | 0.5 N × 4 thrusters | 0.1 N × 4 thrusters |
| Propellant Mass (kg) | ~3.7 | ~0.26 [12] | 0.33 | 0.4 | ~2 [a] [58] |
| Wet Mass (kg) | 7.2 | 1.5 | 1.2 | 10 | 5.55 |
| Total Impulse (N s) | >7290 | 565 | 650 | ~667.08 [a] [57] | >2500 |
| Feed and Pressurization | Pump | PLPS [b]–gas generator | Barrier separated const. pressure gas | Pressurization in orbit–gas generators | Pump |

LFPS: Lunar Flashlight Propulsion System. [a] These values are first-order calculations from available data, exact values were not explicitly indicated by the source. [b] Post Launch Pressure System, a hybrid blow-down gas generator pressurization system [45].

### 5.2. Commercial Off-the-Shelf Monopropellant Thrusters

Commercial off-the-shelf (COTS) components generally receive high interest in the community of small-size spacecraft designers and developers. Recently, COTS green monopropellant thruster belonging to the High-Performance Green Propulsion (HPGP) technology by Bradford ECAPS has acquired special interest to CubeSat propulsion analysts and designers due to their extensive flight heritage and their compelling performance figures. HPGP thrusters were demonstrated and flown in various missions such as in the Mango spacecraft of the PRISMA demonstrator mission by ESA in 2010, in the LEO mission

of STPSaT-5 by the U.S. Government in 2018, and the SkySat LEO imaging constellation by Planet Lab in 2016 to 2020. The HPGP thrusters, Figure 8e, typically operate on the ADN-based LMP-103S green monopropellant and are available in thrust range from 0.1 up to 200 N. Of course, the thrust levels of interest to a CubeSat designer would typically be the 0.1, 0.5, and 1 N thrusters; the performance characteristics of such thrusters are shown in Table 13.

**Table 13.** HPGP thrusters: performance and specifications [59].

| Thruster (HPGP) | 0.1 N | 0.5 N | 1 N |
|---|---|---|---|
| Thrust Range | 30–100 mN | 0.12–0.5 N | 0.25–1 N |
| Inlet Pressure Range (MPa) | 0.23–0.45 | 0.2–0.9 | 0.45–2.2 |
| Nozzle $A_e/A_t$ | 100:1 | 100:1 | 100:1 |
| Steady state vacuum $I_{sp}$ (s) | 196–209 | 178–219 | 204–231 |
| MIB * (mNs) | $\leq 5$ | $\leq 35$ | $\leq 70$ |
| OAL ** (mm) | 55 ex. FCV | 155 | 178 |
| Mass (g) | 40 ex. FCV | 180 | 380 |
| Pull-in Voltage (VDC) | $10 \pm 2.5$ | $28 \pm 4$ | $28 \pm 4$ |
| Holding Voltage (VDC) | 3.3 | $10 \pm 1$ | $10 \pm 1$ |
| Reactor Pre-heating Volt (nominal) (VDC) | 9 | 28 | 28 |
| Reactor Pre-heating Power (regulated) (W) | 6.3–8 | 8–10 | 8–10 |

* Minimum Impulse Bit; ** Over All Length.

Busek Co. also develops a family of green monopropellant thrusters, namely BGT-family, with a thrust range from 0.1 to 22 N. These thrusters operate mainly on the HAN-based AF-M315E green monopropellant but are also compatible with other blends of high-performance green monopropellants, as mentioned on the manufacturer website. The BGT-X1 and BGT-X5 are of particular interest to CubeSat designers and offer nominal thrusts of 0.1 and 0.5 N, respectively, refer to Figure 8f. Performance figures and specifications are presented in Table 14.

**Table 14.** Busek BGT thrusters: performance and specifications [60].

| Thruster | BGT-X1 | BGT-X5 |
|---|---|---|
| Thrust (nominal) | 0.1 N | 0.5 N |
| Throttleable Range (mN) | 20–180 | 50–500 |
| Vacuum specific impulse $I_{sp}$ (s) | 214 | 220–225 |
| MIB * (mNs) | <14 | <50 |
| Catalyst Preheat Power (W) | 4.5 | 20 |

* Minimum Impulse Bit.

## 6. Conclusions

MIMPS-G500mN is a green monopropellant propulsion system that was designed for small-size spacecraft in CubeSat architecture. The propulsion system employs a novel autogenously pressurized micro electric pump feed system which is believed, from the author's point of view, to have a great impact on the propulsion system miniaturization and maximizing performance. Such propulsion systems may offer flexibility and adaptability toward the space mission requirements. One of the major capabilities that this system will provide, when compared to other market available monopropellant systems, is the non-degrading thrust performance for almost the whole mission lifetime which is accredited to the unconventional micro electric pump feed cycle that is employed in the proposed novel feed and pressurization system. Another compelling aspect that is solely associated with MIMPS-G design over any available green monopropellant propulsion system is allowing thrust expandability and clustering of several propulsion modules that will work synchronously to fulfill different maneuver requirements for larger size CubeSats. Despite the obvious complexity, such complex systems can be one step closer

toward their realization due to the existence of modern technologies, such as rapid additive manufacturing, advanced materials for space-use (i.e., carbon fiber and high heat resistance superalloys), and most importantly the technological advancements that lead to the availability of affordable suitable microvalves and micro electric pumps. With the existence of advanced onboard computers, real-time onboard control of such multivariable system shall demonstrate technical and operational feasibility. Pump feed and autogenous pressurization are considered unconventional and an undeveloped concept for small-sized spacecraft liquid propulsion, due to their high complexity, especially with green energetic ionic liquid monopropellants. However, this novel approach for propellant feed and tank pressurization can be a drastic change towards high-performance miniaturized spacecraft and small satellites. The MIMPS-G class of propulsion systems is pending the realization phase. The first model, namely MIMPS-G500mN, is currently in a TRL 2–3; prototyping and testing phases are planned for fall 2021.

A short review for state-of-the-art Energetic Ionic Liquid EIL green monopropellants was presented, and a detailed trade-off study was performed to propose the three candidate green monopropellants for MIMPS-G (i.e., ASCENT, LMP-103S, HNP225). The article extensively discussed the system analysis and design methodology as well as the concept of operations of the proposed propulsion system. The preliminary design study was elaborated, and relevant process diagrams and flowcharts were shown to provide more clarification for the reader. Results were quantitatively tabulated and qualitatively assessed, and 3D CAD models and renders were presented within the article body and in the appendix to provide visualization for the reader.

A market survey was made for the state-of-the-art small satellites monopropellant propulsion systems, as well as commercial off-the-shelf green monopropellant thrusters. Specifications and performance characteristics of such propulsion components are mentioned in Tables 13 and 14. These systems are the MPS-135 4U by Aerojet Rocketdyne [54], BGT-X5 by Busek Co. [43], EPSS C1 by NanoAvionics [53], Pinot-G by IHI Aerospace Co. [57], and the Lunar Flashlight Propulsion System (LFPS) by NASA and Georgia Tech [58]; more detailed discussion about the feed and pressurization in these propulsion systems can be found in the reference [9]. All these systems are in CubeSat standard size, except for the Pinot-G, which was specified in cylindrical dimensions of $\varnothing$350–135 mm. As shown in the table, the pump-fed systems possess the highest total impulse performance values of >7290 and >2500 N s for the MPS-135 and the LFPS, respectively. Next comes the EPSS C1 and the BGT-X5 with values of 650 and 565 N s, which is still a reasonable value for the size of a 1U propulsion system, especially when having a reasonable wet mass to dry mass fraction. On the other hand, the Pinot-G delivers a decent total impulse value, however, the dry mass of the system is surprisingly high, ~9.6 kg compared to 3.5 kg for the MPS-135 4U. The results of this brief analysis along with the performance data and specifications in Table 12 may be considered as reference figures for the MIMPS-G500mN CubeSat green monopropellant propulsion system preliminary design results, as well as to the readers of this article.

**Supplementary Materials:** The following are available online at https://www.mdpi.com/article/10.3390/aerospace8060169/s1. Video S1: MIMPS-G500mN 3D CAD realistic render display.

**Author Contributions:** Conceptualization, A.E.S.N., A.C. and A.P.; methodology, investigation, software, data curation, writing—original draft preparation, A.E.S.N.; supervision, writing—review and editing, A.P. and A.C. All authors have read and agreed to the published version of the manuscript.

**Funding:** This research received no external funding.

**Data Availability Statement:** Any new data created or analyzed were mentioned explicitly within the article and simulation conditions were illustrated.

**Acknowledgments:** I would like to acknowledge Shinji IGARASHI and Yoshiki MATSUURA from IHI Aerospace Co., Ltd. for providing us with information relating to HNP green monopropellants.

**Conflicts of Interest:** The authors declare no conflict of interest.

## Appendix A

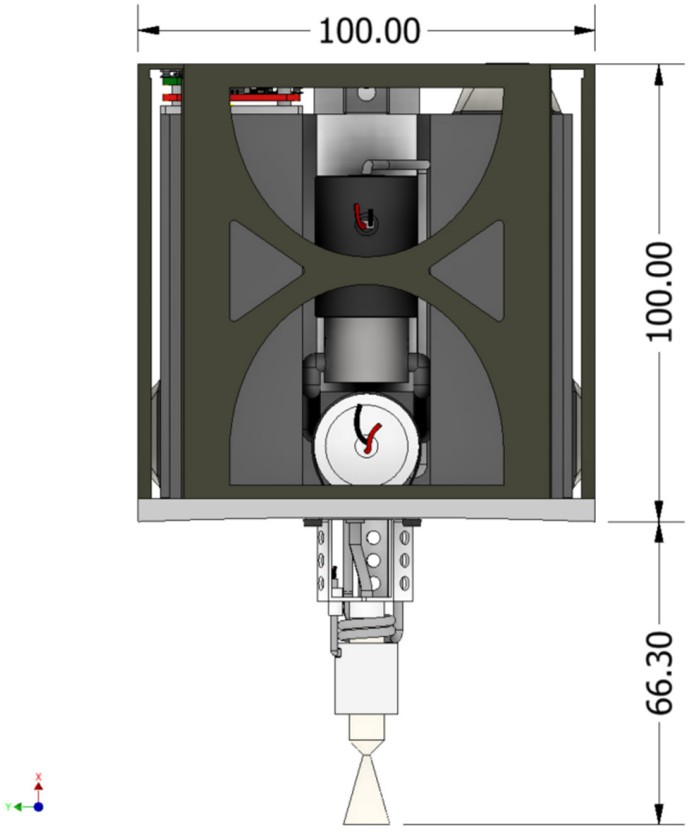

**Figure A1.** MIMPS-G500mN envelope dimensions (mm).

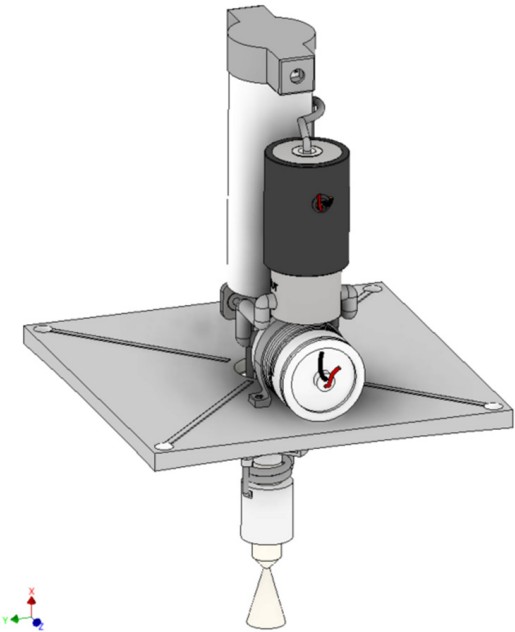

**Figure A2.** Feed and pressurization system (Micro electric pump feed cycle–Autogenous pressurization).

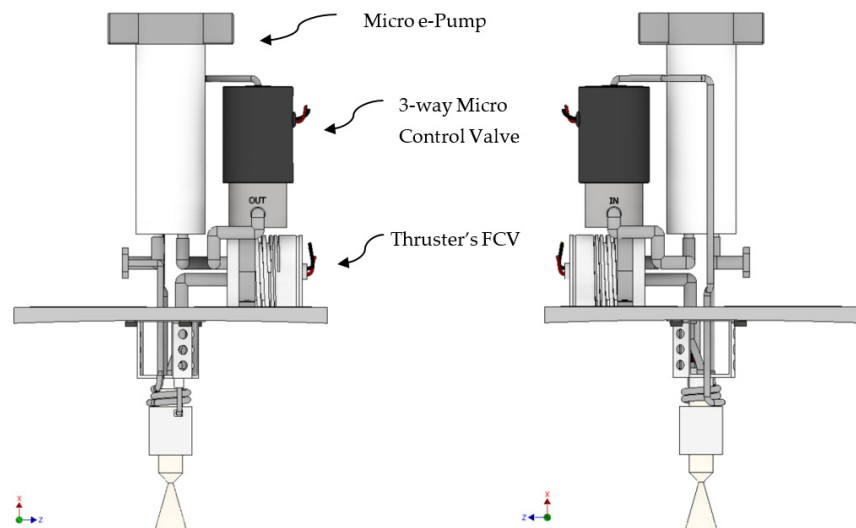

**Figure A3.** Feed and pressurization system side view and components.

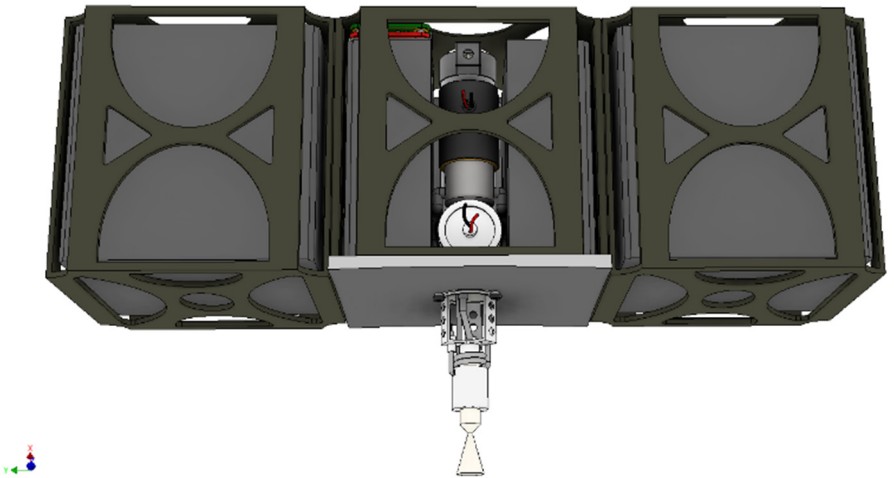

**Figure A4.** MIMPS-G500mN on a 9U CubeSat with two extension side tanks. $I_{tot}$~5000 N s using ASCENT (formerly AF-M315E).

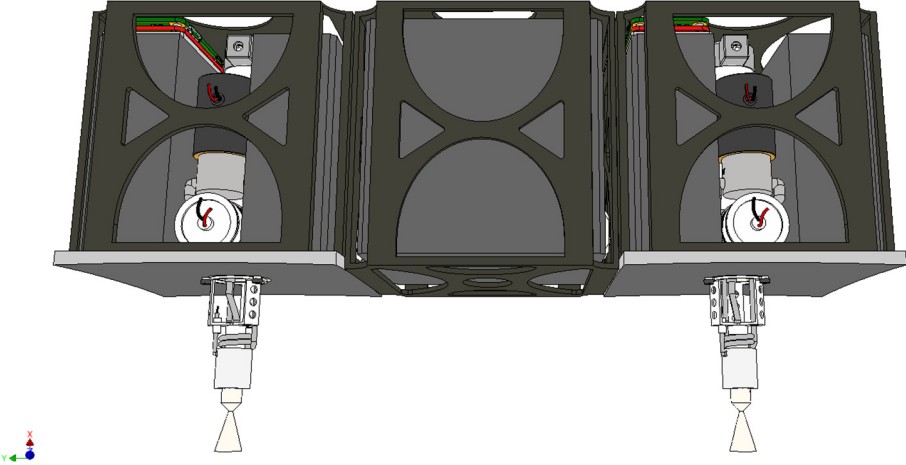

**Figure A5.** MIMPS-G500mN on a 9U CubeSat, 1 N Thrust, and one extension tank. $I_{tot}$~4500 N s using ASCENT (formerly AF-M315E).

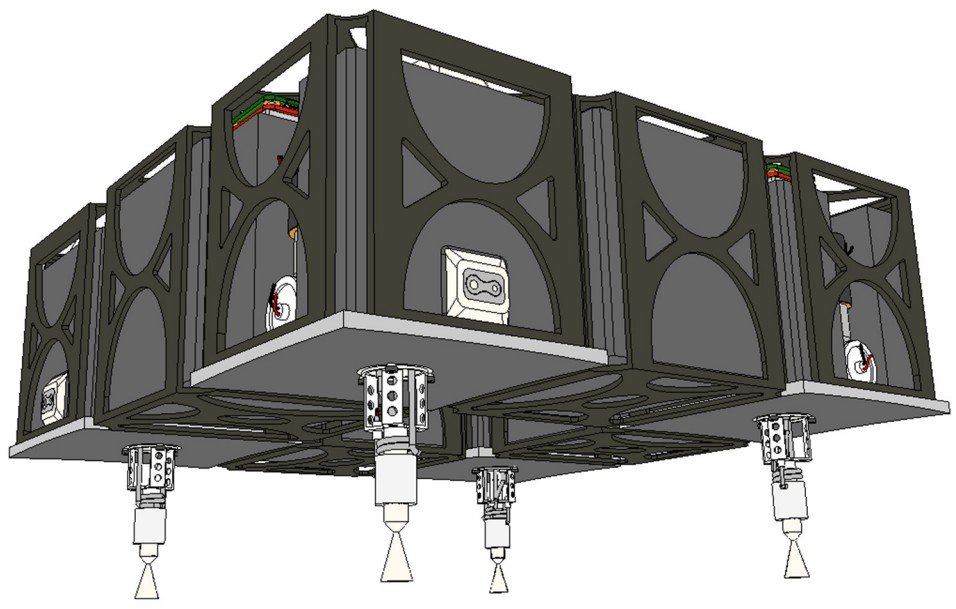

**Figure A6.** MIMPS-G500mN Cluster on a 27U CubeSat, 2 N Thrust, and four extension tanks.

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
