# Peer review of "Modular Impulsive Green Monopropellant Propulsion System (MIMPS-G): For CubeSats in LEO and to the Moon"

_aerospace, doi:10.3390/aerospace8060169_

Round 1
Reviewer 1 Report
Please see report attached.

Author Response
Dear Reviewer,
Thank you for your intensive review and clear comments.
Please find attached our response.
Best Regards,
The Authors

Reviewer 2 Report
This work describes the design of a green energetic ionic liquid propulsion system for small satellites (MIMPS-G500mN). There are really three different topics covered in this work: (1) the MIMPS design including the design rationale and process and flow chart, (2) a review of energetic ionic liquid propellants, and (3) a review of existing green propellant small sat propulsion thrusters/systems. All 3 topics are quite relevant and timely given the community interest and research activities. If this reviewer can make a suggestion, I think ideally, the work would be broken up into two manuscripts: (1) MIMPS propulsion system design, followed by testing, and test results, actual experimental measurements and comparison with the design prediction; (2) Review of existing green EILs/propellants and the thrusters/systems that use them. While this would be an improvement I believe, the manuscript as written makes a new and important contribution and can be published.
Specific comments:
FIG 6 reminded this reviewer of another recent article he read about operating multiple thruster types from one propellant tank (in your case, vapor auxiliary thrusters (which could be heated resistojet or warm gas thruster) and a chemical monoprop thruster): "Rovey, J. L., Lyne, C. T., Mundahl, A. J., Rasmont, N., Glascock, M. S., Wainwright, M. J., Berg, S. P., "Review of multimode space propulsion," Progress in Aerospace Sciences, Vol. 118, pp. 100627, 2020.", which also discusses how emerging EILs can be used for multimode propulsion.
Author Response

(The authors gave the same response as above.)
